# Comparison of Alternative Splicing Landscapes Revealed by Long-Read Sequencing in Hepatocyte-Derived HepG2 and Huh7 Cultured Cells and Human Liver Tissue

**DOI:** 10.3390/biology12121494

**Published:** 2023-12-06

**Authors:** Anna Kozlova, Elizaveta Sarygina, Kseniia Deinichenko, Sergey Radko, Konstantin Ptitsyn, Svetlana Khmeleva, Leonid Kurbatov, Pavel Spirin, Vladimir Prassolov, Ekaterina Ilgisonis, Andrey Lisitsa, Elena Ponomarenko

**Affiliations:** 1Institute of Biomedical Chemistry, Pogodinskaya Street 10, 119121 Moscow, Russiaradkos@yandex.ru (S.R.);; 2Department of Cancer Cell Biology, Engelhardt Institute of Molecular Biology, Russian Academy of Sciences, Vavilova 32, 119991 Moscow, Russia; discipline82@mail.ru (P.S.); prassolov45@mail.ru (V.P.)

**Keywords:** transcriptome, long-read sequencing, alternative splicing, degree of alternative splicing, splice variants abundance, human liver tissue, HepG2 and Huh7 cells, biological pathways, tissue specificity index

## Abstract

**Simple Summary:**

In this study, we evaluated the differences in the alternative splicing (AS) profiles between normal liver tissue, HepG2 malignant cells, and Huh7 malignant cells using a description of AS profiles as arrays of genes characterized by the degree of AS (defined as the number of detected splice variants per gene). In brief, we demonstrated that this new metric can be employed to successfully identify biological pathways that are influenced by the alterations in AS, thereby utilizing a mathematical algorithm previously developed for gene enrichment analysis based on gene expression profiles. Furthermore, since long-read RNA sequencing allows one to also describe the AS profiles as arrays of quantified single transcript isoforms, we employed Yanai’s tissue specificity index (suggested for gene expression analysis) to select groups of genes expressing only one or two splice variants specifically in liver tissue, HepG2 malignant cells, and Huh7 malignant cells, thus providing additional information to that derived from the analysis of gene expression profiles alone. The most of these splice variants were translated into protein products that can contribute to phenotypes of normal and malignant human hepatocytes, thereby making them of interest for the further studying of the mechanisms underlying cell malignization.

**Abstract:**

The long-read RNA sequencing developed by Oxford Nanopore Technologies provides a direct quantification of transcript isoforms, thereby making it possible to present alternative splicing (AS) profiles as arrays of single splice variants with different abundances. Additionally, AS profiles can be presented as arrays of genes characterized by the degree of alternative splicing (the DAS—the number of detected splice variants per gene). Here, we successfully utilized the DAS to reveal biological pathways influenced by the alterations in AS in human liver tissue and the hepatocyte-derived malignant cell lines HepG2 and Huh7, thus employing the mathematical algorithm of gene set enrichment analysis. Furthermore, analysis of the AS profiles as abundances of single splice variants by using the graded tissue specificity index τ provided the selection of the groups of genes expressing particular splice variants specifically in liver tissue, HepG2 cells, and Huh7 cells. The majority of these splice variants were translated into proteins products and appeal to be in focus regarding further insights into the mechanisms underlying cell malignization. The used metrics are intrinsically suitable for transcriptome-wide AS profiling using long-read sequencing.

## 1. Introduction

Alternative splicing (AS) allows for a single gene to be transcribed into two or more mRNA transcripts (splice variants or isoforms), thus ultimately providing a remarkable increase in proteome diversity in higher eukaryotes. The switching via AS to different transcript isoforms is involved in cellular differentiation, the control of cell functions, and the cell response to environmental changes [1,2]. AS is highly regulated, and aberrant splicing contributes to various diseases, including cancer. In humans, over 90% of transcripts undergo alternative RNA processing, and about 15% of hereditary diseases and cancers are thought to be associated with a dysregulation of AS [1,2].

The transcriptome-wide analysis of AS was greatly boosted by the advance of the next-generation sequencing and is mostly based on the high-throughput sequencing of short cDNA fragments (RNA-seq) [3]. Yet, when accurately quantifying gene expression, short-read sequencing in general fails to correctly identify the isoform from which the read originates, since the isoforms from the same gene are similar to a large extent [4,5]. To overcome this issue, two metrics have been suggested for measuring AS events at the transcriptome-wide level—‘exon usage’ [6] and the PSI (percent spliced in) index [7]. Both metrics indicate, in fact, how frequently a given exon is included into the transcript isoforms of a corresponding gene and can be calculated directly from the row read counts, hence avoiding uncertainties regarding the short-read assembly to reveal a splicing pattern. Despite the ongoing attempts to improve bioinformatics tools for RNA-seq-based assembly to quantify splice variants (e.g., [8,9]), it still remains quantitatively challenging.

The emergence of third-generation sequencers such as those of Oxford Nanopore Technologies (ONT) has allowed for sequencing RNA or cDNA as a single molecule, thus providing long reads, which can span multiple exons. Long-read sequencing significantly simplifies the detection of transcript isoforms, thus directly revealing splicing patterns [2]. This makes the normalized abundance of single (individual) transcript isoforms (rather than the gene expression measured as an integral normalized abundance of transcript isoforms ascribed to the gene) the more appropriate metric for the analysis of AS profiles in the case of long-read ONT sequencing than the ‘exon usage’ or PSI index. Indeed, though the ‘exon usage’ or PSI index continue to be used for AS profiling based on long-read sequencing data (e.g., [10,11,12]), the description of AS profiles in terms of the abundance of single isoforms has also been utilized in ONT-based transcriptome-wide studies (e.g., [13,14,15]). On the other hand, as we recently suggested [16], AS profiles can be described regardless of a particular expression of a given transcript isoform as arrays of genes, where each gene is characterized by the number of detected splice variants ascribed to that gene (here referred to as the ‘degree of alternative splicing’, or the DAS).

The aim of this study was to further explore the utility of such metrics as the DAS or abundances (in transcripts per million, or TPM) of single transcript isoforms for revealing the differences in AS profiles between various cell/tissue types (which we further refer to as ‘phenotypes’ for convenience) using long-read sequencing datasets. We employed bioinformatics tools that were previously developed for gene expression analysis, such as GSEA (gene set enrichment analysis) [17] and the graded tissue specificity index τ [18]. These tools were commonly applied to identify the biological pathways that are influenced by differential gene expression (e.g., [19,20] and references therein) or to find tissue-specific signatures of gene expression [21,22,23]. Samples of human liver tissue and hepatocyte-derived HepG2 and Huh7 cultured cells were used for this purpose, and the extracted mRNA was subjected to long-read ONT sequencing. HepG2 and Huh7 are cell lines derived from hepatoblastoma and hepatocellular carcinoma tumor tissues, respectively [24,25], which are widely used as models in biotransformation studies (e.g., [26,27,28]) or for studying the processes associated with the malignant transformation of hepatocytes ([29,30,31,32], to mention a few). Among these cell lines, long-read nanopore sequencing was only applied to AS profiling in HepG2 cells, which focused either on a particular group of genes such as cytochrome P450 genes [33] or on nonpolyadenylated transcripts [34].

## 2. Materials and Methods

### 2.1. Cell Lines and Liver Tissue

Samples of human liver were collected at time of autopsy from 3 male donors aged 65, 38, and 54 years (designated further as donors 1, 2, and 3, respectively) with the approval of the N.I. Pirogov Russian State Medical University Ethical Committee (protocol #3; 15 March 2018) and the informed consent from donor’s representatives. The donors were HIV and hepatitis free, and the sections had no histological signs of liver diseases. The postmortem resected samples were immediately placed into RNAlater RNA Stabilization Solution (Thermo Fisher Scientific, Waltham, MA, USA) and stored at −20 °C until further use.

The HepG2 and Huh7 cells were purchased from Merck (Darmstadt, Germany) and Thermo Fisher Scientific, respectively. Cells were cultivated in a DMEM growth medium supplemented with 10% fetal bovine serum and 100 units/mL penicillin/streptomycin (all from Dia-M, Moscow, Russia) in a humidified atmosphere with 5% CO_2_ at 37 °C to ≈80% confluence. Afterwards, cells were harvested, washed with phosphate buffered saline (Dia-M, Moscow, Russia), snap frozen in liquid nitrogen, and stored in liquid nitrogen vapor till further use.

### 2.2. RNA Isolation, Library Preparation, and Long-Read Sequencing

To prepare sequencing libraries, total RNA was isolated from cells or tissue samples with an RNeasy Mini Kit (Qiagen, Hilden, Germany) according to the manufacturer’s manual, and it was quantified using a NanoDrop-1000 spectrophotometer (Thermo Fisher Scientific); its quality was assessed using a Bioanalyzer 2100 System (Agilent Technologies, Palo Alto, CA, USA). The RNA integrity numbers were 7.8 or higher for all RNA preparations. The mRNA extraction was conducted with a Dynabeads™ mRNA Purification Kit (Thermo Fisher Scientific) following the manufacturer instructions. The mRNA was quantified using a Qubit 4 fluorometer and a Qubit RNA HS Assay Kit (Thermo Fisher Scientific). The mRNA preparations were either used immediately or frozen at −80 °C for short-term storage.

The sequencing libraries were prepared with a Direct RNA Sequencing Kit (SQK-RNA002, ONT, Oxford, UK) strictly following the manufacturer’s protocol. The long-read sequencing was carried out on a MinION nanopore sequencer (ONT) in 48 h single runs using FLO-MIN106 flow cells. The row data were processed using the guppy_basecaller 3.1.5 software (ONT) as described in [35]. During processing, the data were filtered using the guppy_basecaller software with a quality score parameter > 7.0. The quality control of reads was performed with the MinIONQC.R script [36]. Mapping was carried out with the minimap2 v.2.17 software [37] using the Gencode38 genome assembly (release GRCh40). Overall, mRNA from 11 biosamples was sequenced (5 samples of HepG2 cells, 3 samples of Huh7 cells, and a sample of liver tissue from each of the 3 donors). The number of mapped reads for each biosample sequenced is presented in Appendix A. To account for differences in sequencing depth, the sequencing outputs were adjusted to the minimal output, which was received for the sample of liver tissue of donor 3 (Appendix A) with the Picard DownsampleSam tool (https://broadinstitute.github.io/picard/, accessed on 10 September 2023). Transcript abundance was quantified in TPM with the Salmon 0.12/1.1.0 software [38] employing the Salmon Quant tool. The sequencing data were deposited to the NCBI Sequence Read Archive (PRJNA765908, PRJNA893571, and PRJNA635536).

### 2.3. Data Analysis

The Salmon Quant output files with the evaluated abundances of splice variants were assembled into a common Excel table, and then the table was truncated so as to leave only records corresponding to splice variants expressed by protein-coding genes in at least one of the biosamples tested (presented as Appendix A in the SMs). The table contains 13 columns (columns of ENSG and ENST identifiers, as well as 11 columns with TPM values of transcripts isoforms for each biosample) and a long list of rows with ENST identifiers (the overall number of identifiers was 30556). The table was also converted into another table (presented as Appendix A), where each gene (14427 ENSG identifiers) was characterized by its DAS value (the number of splice variants assigned to that gene) for each biosample tested.

Appendix A was used to calculate the value of τ index as follows:(1)$$\tau =\frac{\sum_{i=1}^{N}\left(1-Xi\right)}{N-1},$$
where *N* = 3 is the number of phenotypes tested (*viz.*, liver tissue, HepG2 cells, and Huh7 cells), and *X*i is the expression profile component normalized by the maximum component value among the phenotypes [18]. We used splice variants abundances in TPM, which were averaged over all biosamples of a given phenotype, as the expression profile components. Alongside calculating τ index for splice variants, we also calculated it for gene expression. In this case, the gene’s expression profile component was a sum of abundances of all splice variants corresponding to the gene. The values of τ index are presented in Appendix A.

The data in Appendix A were used to construct input matrixes for a pairwise comparison of 3 phenotypes with the GSEA software v. 4.3.2, which was downloaded from https://www.gsea-msigdb.org/gsea/index.jsp (accessed on 10 September 2023). The pairwise comparisons were carried out using pathway databases WikiPathways [39], Reactome [40], KEGG [41], and BioCarta [42] in a gene set permutation mode following GSEA settings: «The number of permutations» == 1000, «Permutation type» == gene_sets. We used datasets «as is» in the original format as an expression (in terms of the number of splice variants per gene) matrix. The normalized enrichment score (NES) was calculated using the GSEA software with the FDR < 0.25. |NES| > 1.5 and *p* < 0.05 were set to define significantly enriched pathways.

## 3. Results

### 3.1. Biological Pathways Influenced by Differences in Alternative Splicing in Liver Tissue and Hepatocyte-Derived Cell Lines

Since a degree of alternative splicing can be easily extracted for each gene from the output of long-read ONT sequencing (Appendix A), we tested our datasets for groups of genes whose splicing might be systematically altered among liver tissue and hepatocyte-derived cell lines. We employed GSEA, which combines information from the members of previously defined sets of genes (*viz.*, involved in particular biological pathways) to increase the signal relative to the noise and to improve the statistical power [17]. Instead of the overall abundance of transcript isoforms being expressed by a gene, which is ordinarily utilized in GSEA to characterize gene expression, we used another metric, the number of splice variants per gene, and coined such an analysis here as ‘Splicing-based Pathway Enrichment Analysis’ (SPEA) to account for the nonstandard type of input data. In the conventional GSEA, the positive value of the NES indicates that the genes in the analyzed dataset, which are involved in the corresponding pathway, exhibit, on average, a higher expression in phenotype 1 compared to that in phenotype 2, which was taken as its counterpart. Otherwise, if the NES value is negative, that indicates a lower expression of these genes on average. In the case of SPEA, the positive and negative values of the NES would indicate that the genes are, on average, characterized by the higher or lesser degrees of alternative splicing, respectively.

The results of the pairwise comparisons of the AS profiles described in terms of the DAS values are presented in Figure 1. To increase the stringency of our analysis, we included in the SPEA only the genes that demonstrated a ‘stable’ expression—transcripts related to such genes were detected in all the biosamples tested. As is seen from Figure 1, the patterns of the revealed biological pathways were quite different for the phenotype pairs. Nonetheless, there is some similarity when the liver tissue is compared with the HepG2 or Huh7 cells. In both cases, the genes involved in the complement cascade and peroxisome proliferator activated receptor signaling pathways, as well as the number of metabolic pathways, were characterized, on average, by a higher degree of alternative splicing in the liver tissue compared to the cell lines. In contrast, the genes characterized by lower DAS values, on average, were involved in translation and translation-related pathways, as well as in pathways related to mRNA maturation and alternative splicing (Figure 1). However, the genes involved in plasma lipoprotein biosynthesis were characterized by higher DAS values in the liver tissue compared to the HepG2 cells, but not in the Huh7 cells (at least in terms of statistically significant differences). In addition, the same pattern was revealed for the pharmacogenes (genes involved in the ‘Phase II conjugation of compound’ and ‘Drug metabolism cytochrome P450′ pathways).

Compared to the liver tissue, the hepatocyte-derived malignant cells differed between themselves to a lesser degree with regard to the biological pathways influenced by their AS profiles. Nonetheless, the HepG2 cells showed a higher degree of alternative splicing on average for the genes involved in cholesterol biosynthesis, DNA repair, and oncogenic-related signal pathways, while the Huh7 cells showed a lesser degree of alternative splicing for the genes involved in transcription-related pathways (Figure 1).

### 3.2. Genes with a Phenotype-Specific Splice Variant or Integral Expression

The τ index was suggested by Yanai et al. [18] as a quantitative, graded scalar measure of the specificity of expression for a given gene across various tissues. The index ranges from 0 to 1, and values of τ < 0.15 represent genes whose expressions are rather similar across the tissues of interest (e.g., housekeeping genes), whereas τ > 0.85 indicates genes that are preferentially expressed in one of these tissues. They were referred to as those with the one-tissue-specific expression [18]. Here, we are referring to them as ‘phenotype-specific’ for the sake of convention.

Unlike the original work of Yanai et al. [18], we calculated τ index values not only for the genes, but also for each and every splice variant detected. The distributions of the number of spice variants and the genes according to the value of the τ index over the range from 0 to 1 are presented in Figure 2. The substantial number of genes fell into the category of ‘phenotype-specific’—4263 out of 14,427 (Appendix A vs. Appendix A), or about 30%. In the case of the splice variants, the proportion markedly increased—by over 50%, or 15,491 out of 30,556 (Appendix A vs. Appendix A)—thus indicating that the expression of single splice variants exhibited a much greater specificity between the phenotypes tested compared to that of the genes. Interestingly, the portion of genes characterized by high values of the DAS decreased in the subsets of genes with either phenotype-specific integral expression or phenotype-specific expression of the single splice variants (Appendix A).

We restricted further analysis to the genes and splice variants with the highest value of the τ index of one, which we regarded as the most phenotype-specific. To tighten our analysis, we also took into consideration only the genes and splice variants that were stably expressed in a given phenotype: here, that means the genes or splice variants were detected in all the biosamples of the given phenotype. Figure 3 presents Venn diagrams built for these genes and splice variants for liver tissue, HepG2, and Huh7 cells. As can be seen, the subsets of genes overlapped, with some genes exhibiting the phenotype-specific integral expression and other genes only exhibiting the phenotype-specific expression for particular splice variants. Since the expression of all these genes can be considered as phenotype-specific one way or another, we combined the gene subsets for each phenotype tested (Figure 3) and subjected the combined sets of genes (164, 237, and 379 genes for HepG2 cells, Huh7 cells, and liver tissue, respectively) to gene ontology (GO) analysis.

Figure 4 shows the top 10 biological pathways revealed by the GO analysis for the sets of genes with phenotype-specific expression. Among these pathways, the pathways related to cell malignization dominated in the pathways list for hepatocyte-derived malignant cells (‘RHO GTPase cycle’, ‘p53 downstream pathway’, ‘regulation of MAPK cascade’, and ‘signaling by BRAF and RAF1 fusion’ for HepG2 cells and ‘transcriptional misregulation in cancer’, ‘cell fate commitment’, and, to some extent, ‘response to UV’ for Huh7 cells). For the liver tissue, the genes with phenotype-specific expression were found to be involved in pathways that are characteristic of the production of plasma blood proteins and immune-related processes (Figure 4).

As can be seen from Figure 3, there were subsets of genes that only showed the phenotype-specific expression for particular splice variants (90, 89, and 99 genes for liver tissue, HepG2 cells, and Huh7 cells, respectively). The overwhelming majority of these genes expressed a single splice variant. Nevertheless, eight of them expressed two splice variants (listed in Table 1), and no genes expressed three or more. For the Huh7 and HepG2 cells, only one splice variant for each of these genes was translated into a protein, while the second splice variant gave no corresponding proteoform detected to date. In the case of the liver tissue, both of the splice variants of the SLC17A1 gene were translated into the same protein (solute carrier family 17 member 1), whereas the splice variants of the APOC3 gene (encoding apolipoprotein C3), revealed as phenotype-specific, gave no protein products (Table 1).

We further attempted to gain insights into whether the genes with strictly phenotype-specific single splice variant expression produced any known protein products. For that, we randomly selected 10 genes in the corresponding gene sets (for the liver tissue, HepG2 cells, and Huh7 cells) to serve as their representative samples. The lists of the selected genes are provided in Appendix A. The analysis of the proteoforms encoded by those genes demonstrated that the proteoforms have a canonic, noncanonic, or predicted status for correspondingly eight, one, and one genes in the liver tissue, respectively, for seven, one, and two genes in the Huh7 cells, respectively, and for four, two, and four genes in the HepG2 cells, respectively (Appendix A). Thus, it appears that, in each phenotype tested, more than a half of the genes with strictly phenotype-specific single splice variant expression produced detected proteins.

## 4. Discussion

Alterations to AS can markedly influence cell identity (cell phenotype), and different approaches were undertaken to gain insights into AS-related events. Considerable efforts have been put into finding novel transcripts isoforms, which can potentially contribute to cell phenotype (e.g., [43] and references therein). However, the cell phenotype can also be influenced by differences in AS profiles via a variation in the composition of the known transcript isoforms. To describe AS profiles, metrics such as ‘exone usage’ or the PSI index are commonly employed [3,6,7,44]. Yet, they can rather be considered as substitutes for true AS profiling, and their wide use in the short-read RNA-seq is stipulated by ambiguities in the identification and quantification of different transcript isoforms that are originated from the same gene. Though these metrics continue to be used in the analysis of long-read ONT sequencing data (e.g., [10,11,12]), AS profiles are also described in terms of the expression of single transcript isoforms (as opposed to the gene expression, which is an integral expression of all the detected transcript isoforms assigned to the gene) [13,14,15], since, in this case, the standard treatment of raw sequencing data directly provides the splice variant abundances for each and every gene. Here, we utilized both approaches—by presenting AS profiles either as an array of all the transcript isoforms with quantified abundances or as an array of the genes with the corresponding DAS values—and applied them to differentiate the AS in three types of biosamples: from normal human liver tissue and from two hepatocyte-derived malignant cell lines.

Presently, there is a variety of bioinformatics tools aimed at the analysis of gene expression profiles [45]. Some of them, like GSEA, have been suggested to gain insights into the biological mechanisms that can be affected by differential gene expression [17]. Since the GSEA computational algorithm is based on ranking genes according to their quantitative characteristics (in the conventional GSEA, this is in regard to the differential expression in terms of the relative transcript abundances), it appears reasonable to utilize this algorithm to gain insights into the biological processes influenced by differential splicing by simply ranking genes according to their differential degrees of alternative splicing. Indeed, the pairwise comparisons using SPEA (as we called GSEA with the nonstandard type of input data) between liver tissue, HepG2 cells, and Huh7 cells made it possible to reveal the biological pathways that are apparently influenced by differential splicing (Figure 1). The revealed biological pathways reasonably agree with the phenotypes of the tested biosamples. Thus, the biological pathways identified for liver tissue (such as the cascade of the complement and coagulation system, the signaling pathways of the receptor genes activated by peroxisome proliferators PPARs, and the metabolism of amino acids, vitamins, and fatty acids) include genes whose protein products are expressed in liver tissue and are involved in the realization of basic liver functions [46,47,48]. In contrast, the hepatocyte-derived malignant cell lines HepG2 and Huh7 were generally characterized by a reduced expression regarding the genes involved in the complement system, PPAR signaling, and fatty acid metabolism pathways [49]. An increase in the number of splice variants for the genes involved in these pathways in liver tissues may well be associated with their increased integral expression [50] rather than with an increase in the expression of a particular splice form. Among the pathways whose genes were characterized by an increase in the number of splice variants in the liver tissues in comparison to the Huh7 cells, processes related to drug metabolism were also identified, namely, the phase II of biotransformation and drug metabolism by cytochromes of the P450 family (Figure 1), that may also be associated with the reduced expression of the pharmacogenes in the cell lines [51].

An increase in the number of splice variants in the Huh7 and HepG2 cells for pathways associated with apoptosis, translation, mRNA maturation, and alternative splicing may be attributed to the cancerous origins of the analyzed cell lines [52]. Alterations in the AS events are known to occur in tumors for genes involved into these biological processes, thereby leading to the formation of tumor-specific splice variants and dramatically changing, among others, their resistance to chemotherapy [53].

Alongside the DAS as a metric to describe AS profiles in terms of the number of splice variants per gene, we also described these profiles as arrays of splice variants that were characterized by their abundance in TPM. By employing the graded tissue specificity index—the τ index [18]—we attempted to explore whether such a description of AS profiles can provide additional information to that provided by the gene expression profiles with the regard to the tested phenotypes. The τ index-based analysis of the AS profiles did reveal subsets of genes exhibiting a strictly phenotype-specific expression of single or two splice variants, thereby supplementing the subsets of genes with phenotype-specific integral expression (Figure 3). Most of these splice variants (apparently not less than 60%) were translated into proteins (Table 1 and Appendix A), thus apparently contributing to the phenotypes of the tested biosamples.

Clearly, the AS profiles as arrays of splice variants differentiate HepG2 cells, Huh7 cells, and liver tissue to a larger extent that the gene expression profiles. Indeed, over 50% of the splice variants demonstrated phenotype-specific expression compared to about 30% of the genes (Figure 2 and Appendix A). The same held true for the sizes of subsets of strictly phenotype-specific genes and splice variants (τ = 1): they related as 53%, 65%, and 85% for HepG2 cells, Huh7 cells, and liver tissue, respectively (Figure 3). The phenotype-specific expression of the splice variants reasonably agrees with the known fact that the number of AS events in tumors and normal tissues significantly differs [54,55]. Furthermore, for the hepatocyte-derived HepG2 and Huh7 cells, we also found a substantial number of splice variants whose expressions strictly differentiated these cells. That may reflect different origins for these malignant cells. Indeed, the HepG2 cell line was established from hepatoblastoma cells, which are a result of the malignant transformation of pluripotent hepatic stem cells and may, to some extent, retain the phenotype of blast cells [56]. With respect to the Huh7 cell line, it was established from hepatocellular carcinoma (HCC) resulting from the malignant transformation of differentiated hepatocytes [24,25,56]. Hepatoblastoma cells are known to lack the expression of the proteins that are characteristic for HCC cells [57]. It is worth noting though that, even in the case of cell lines that have originated from different tumors of the same type such as HCC, differences in the gene expression profiles have also been observed [49].

The GO analysis of the combined sets of genes with strictly phenotype-specific expression (either integral or of a particular splice variant or variants) revealed that these genes are involved in different biological pathways. Thus, for HepG2, the phenotype-specifically expressed genes are involved in processes associated with the proliferation and migration of tumor cells, whereas for Huh7, the phenotype-specifically expressed genes are involved in the dysregulation of gene transcription and intracellular transport (Figure 3). Increased cell proliferation and migration are both characteristic of poorly differentiated blast cells and cancer cells. Moreover, the altered activity of RHO GTPases can lead to cancer progression [58]. The detected specific genes and splice variants related to the RHO GTPase signaling pathway render themselves to further investigation as potential markers for the differential diagnosis of blastoma and carcinoma cells.

## 5. Conclusions

The degree of alternative splicing defined as the number of splice variants per gene can be used in gene enrichment analysis as a quantitative characteristic of AS, thereby allowing the mathematical algorithm developed for the analysis based on gene expression profiles to be applied to the analysis of AS profiles revealed by long-read ONT sequencing. Contrary to metrics such as the ‘exon usage’ or PSI index, the degree of alternative splicing is easily derived from the mapping output in long-read sequencing, without cumbersome calculations. This metric appears to be an intrinsically suitable metric for evaluating the impact of alterations in AS on the biological pathways in normal and malignant cells, as well as in malignant cells with different origins. Furthermore, the described AS profiles in terms of the abundance of single splice variants appear to be beneficial and add to the information that can be derived from gene expression analysis alone. Using the graded tissue specificity index (τ index), we were able to select additional sets of genes expressing one or two splice variants specifically in liver tissue, HepG2 cells, and Huh7 malignant cells. The majority of these splice variants were translated into proteins products and appeal to be the focus of further insights into mechanisms underlying cell malignization.

## Figures and Tables

**Figure 1 biology-12-01494-f001:**
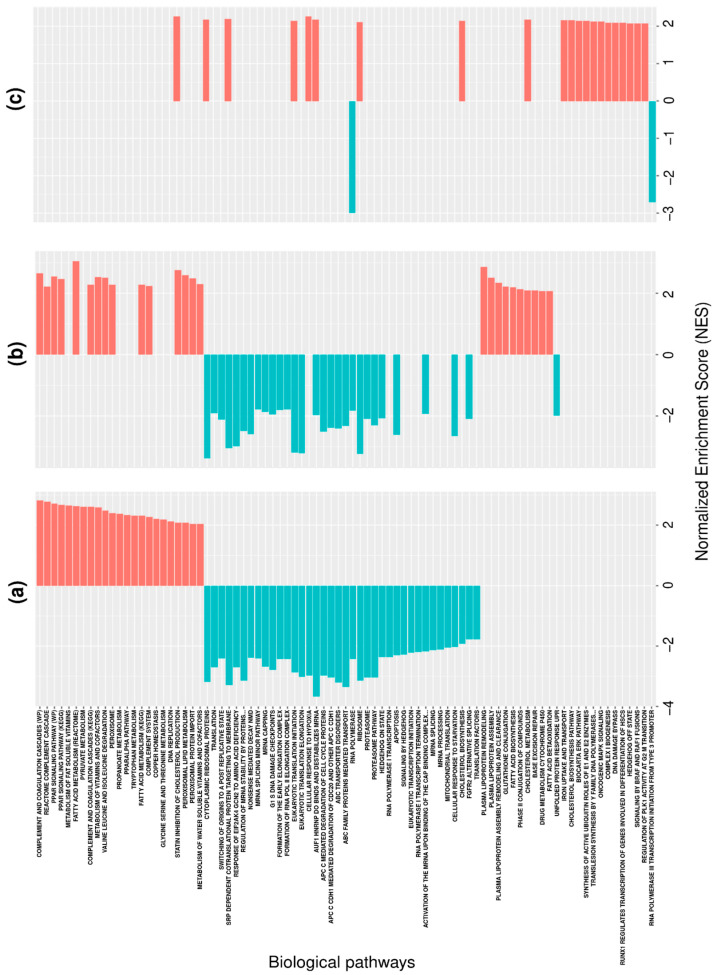
The biological pathways revealed by SPEA via a pairwise comparison of alternative splicing profiles described in terms of degree of alternative splicing. Red and blue colors indicate positive and negative values of NES, respectively. Panels (**a**–**c**) correspond to liver tissue and Huh7 cells, liver tissue and HepG2 cells, and Huh7 and HepG2 cells, taken as phenotypes 1 and 2, respectively.

**Figure 2 biology-12-01494-f002:**
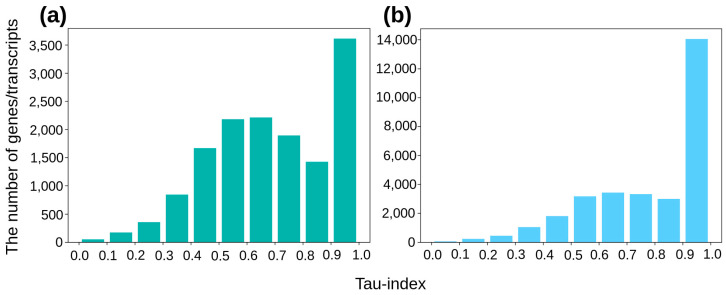
The distributions of the number of genes (**a**) and splice variants (**b**) according to the value of τ index.

**Figure 3 biology-12-01494-f003:**
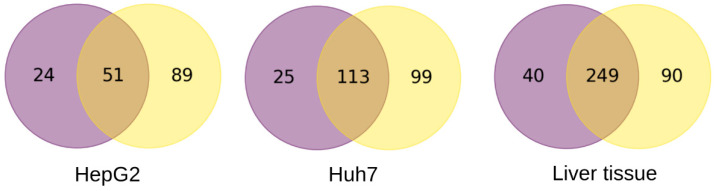
The Venn diagrams for the subsets of genes with the strictly phenotype-specific overall expression (purple circles) and of genes exhibiting the strictly phenotype-specific expression for at least one splice variant (yellow circles) for HepG2 and Huh7 cells and liver tissue. The strictly phenotype-specific expression is characterized by the τ index value of 1.

**Figure 4 biology-12-01494-f004:**
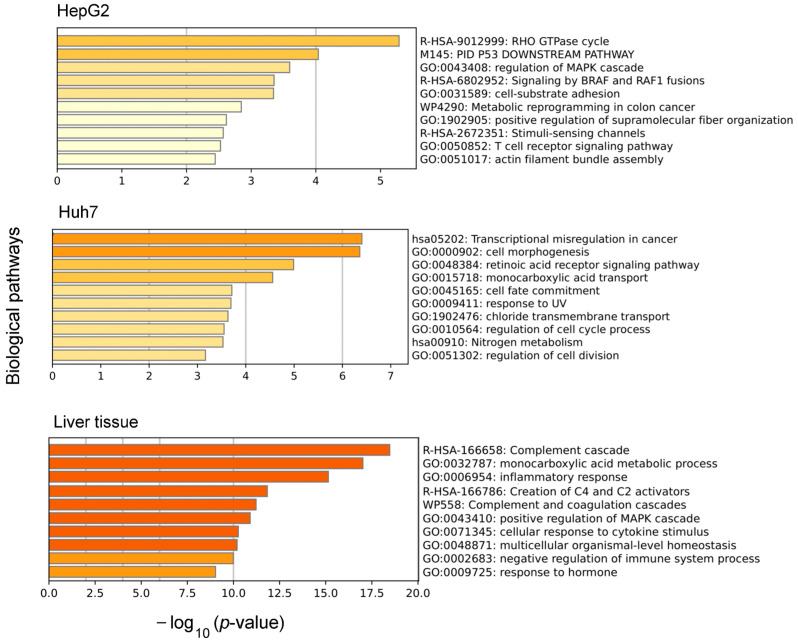
The top-10 biological pathways revealed by gene ontology analysis for the sets of genes with phenotype-specific expression of genes or at least one splice variant for HepG2 and Huh7 cells and liver tissue.

**Table 1 biology-12-01494-t001:** The list of splice variants with phenotype-specific expression and the corresponding proteoforms.

Phenotype	Transcript (Splice Variant) Name	Protein UniProt ID	Current Proteoform Status
Liver tissue	SLC17A1-201	Q14916-1	canonic
Liver tissue	SLC17A1-204	Q14916-1	canonic
Liver tissue	ALDOB-203	P05062	canonic
Liver tissue	ALDOB-207	A0A3B3IS80	predicted
Liver tissue	APOC3-202	B0YIW2	predicted
Liver tissue	APOC3-205	B0YIW2	predicted
Huh7 cells	UROS-211	A0A087WZB7	predicted
Huh7 cells	UROS-203	P10746	canonic
Huh7 cells	GJC1-204	Q5H9P2	predicted
Huh7 cells	GJC1-206	P36383	canonic
HepG2 cells	ASPHD1-203	I3L2A5	predicted
HepG2 cells	ASPHD1-201	Q5U4P2	canonic
HepG2 cells	NEDD4L-205	Q96PU5	canonic
HepG2 cells	NEDD4L-225	K7EKL1	predicted
HepG2 cells	SLC13A3-201	Q8WWT9	canonic
HepG2 cells	SLC13A3-205	C9J4A3	predicted

## Data Availability

The data were deposited into the Sequence Read Archive (https://www.ncbi.nlm.nih.gov/sra (accessed on 27 October 2023)) (PRJNA765908, PRJNA893571 and PRJNA635536).

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
