# Peer review of "Comparison of Alternative Splicing Landscapes Revealed by Long-Read Sequencing in Hepatocyte-Derived HepG2 and Huh7 Cultured Cells and Human Liver Tissue"

_biology, 2023, doi:10.3390/biology12121494_

Round 1

Reviewer 1 Report

Comments and Suggestions for Authors

This study highlights the remarkable potential of long-read RNA sequencing technology from Oxford Nanopore Technologies in advancing our understanding of alternative splicing (AS) in transcriptomes. The ability to directly quantify transcript isoforms and present AS profiles as arrays of single splice variants with varying abundances opens up new avenues for research in this field. The authors demonstrate the utility of describing AS profiles regarding the degree of splicing (DS), which measures the number of splice variants per gene. This approach allows a deeper exploration of how AS impacts biological pathways in human liver tissue and hepatocyte-derived malignant cell lines (HepG2 and Huh7). By employing the Gene Set Enrichment Analysis mathematical algorithm, the authors uncover the specific genes and pathways influenced by AS alterations. In summary, the metrics proposed in this study represent a valuable addition to the existing toolbox for characterizing AS profiles, especially in the context of long-read sequencing. However, the definition of DS does not precisely reflect the alternative splicing and transcriptome diversity. Here are some of my concerns:

Major:

1.       The definition of the “degree of splicing” sounds like the splicing level of a gene from nascent to mature RNAs and it should be the relative ratio of reads from spliced RNAs vs. unspliced RNAs. I think the “degree of alternative splicing” should be more accurate.

2.       Indeed, the number of splice variants per gene can evaluate the complexity of alternative splicing for each gene. However, some genes only have one isoform, and some genes have a lot of annotated isoforms, should we consider the relative ratio between the number of expressed isoforms vs. annotated isoforms for each gene is better? Since different genes have different backgrounds.

Minor:

1.       Please specify y- and y-labels in the main figures.

2.       Change the label of Sheet 1 in Table S2 into English.

3.       Indicating the version of software used in the manuscript.

4.       Upload the codes or scripts for data analysis.

Comments on the Quality of English Language

Please make a minor modification to the language and grammar.

Author Response

This study highlights the remarkable potential of long-read RNA sequencing technology from Oxford Nanopore Technologies in advancing our understanding of alternative splicing (AS) in transcriptomes. The ability to directly quantify transcript isoforms and present AS profiles as arrays of single splice variants with varying abundances opens up new avenues for research in this field. The authors demonstrate the utility of describing AS profiles regarding the degree of splicing (DS), which measures the number of splice variants per gene. This approach allows a deeper exploration of how AS impacts biological pathways in human liver tissue and hepatocyte-derived malignant cell lines (HepG2 and Huh7). By employing the Gene Set Enrichment Analysis mathematical algorithm, the authors uncover the specific genes and pathways influenced by AS alterations. In summary, the metrics proposed in this study represent a valuable addition to the existing toolbox for characterizing AS profiles, especially in the context of long-read sequencing. However, the definition of DS does not precisely reflect the alternative splicing and transcriptome diversity. Here are some of my concerns:

Major:

  1. The definition of the “degree of splicing” sounds like the splicing level of a gene from nascent to mature RNAs and it should be the relative ratio of reads from spliced RNAs vs. unspliced RNAs. I think the “degree of alternative splicing” should be more accurate.

We are thankful to the Reviewer for the positive evaluation of our work in general. As to the definition of the number of splice variants per gene as the “degree of splicing”, we have to agree with the Reviewer that this definition may sound confusing. We have supplemented the definition with the word “alternative” to make it more specific and abbreviated as “DAS” (degree of alternative splicing) throughout the manuscript, following the recommendation of the Reviewer.

  1. Indeed, the number of splice variants per gene can evaluate the complexity of alternative splicing for each gene. However, some genes only have one isoform, and some genes have a lot of annotated isoforms, should we consider the relative ratio between the number of expressed isoforms vs. annotated isoforms for each gene is better? Since different genes have different backgrounds.

The suggestion made by the Reviewer is quite interesting and valuable. The metric defined in the way suggested by the Reviewer does seem to be more appropriate for describing alternative splicing profiles than that simply defined as the number of splice variants per gene. We will undoubtedly evaluate this metric in the future studies on alternative splicing. Yet, at the moment, we would prefer to stay with the definition of the “degree of alternative splicing” as the number of detected splice variants per gene for a sake of continuity with our very recently published paper (ref. 16 of the revised manuscript).

Minor:

  1.       Please specify y- and y-labels in the main figures.

The figure axes have been labeled upon revision. We are thankful to the Reviewer for pointing out this omission and apologize for the inadvertence.

  1.       Change the label of Sheet 1 in Table S2 into English.

The label of Sheet 1 in Table S2 has been corrected. We thank the Reviewer for pointing out this omission.

  1. 3.       Indicating the version of software used in the manuscript.

The versions of the software used have been provided in the revised manuscript.

  1.       Upload the codes or scripts for data analysis.

No own codes or scripts were used for data analysis in the presented work. For the MinIONQC.R script, the relevant reference (ref. 36) has been provided upon revision (line 131).

All changes made in the manuscript upon revision are highlighted with yellow.

Reviewer 2 Report

Comments and Suggestions for Authors

Authors used single splice variants and the degree of splicing as metrics for transcriptome data analysis. Using these metrics, they applied the gene set enrichment analysis to the data for three cell types (phenotypes) and demonstrated that these metrics can be informative to distinguish the transcriptome in these cell types, giving an additional information to the conventional gene expression levels.

I have the following comments/questions:

-       Are there any other attempts to use the same metrics in the literature? The idea to analyze the abundances of splice variants instead of the total number of transcripts for a given gene seems straightforward given the technology, so it's hard to believe that this study is new regarding this aspect. If other attempts of this kind exist, they should be reviewed in the Introduction (which seems too short otherwise), and the novelty of the presented study should be specified more accurately.

-       Is it possible to change the subsection titles in the Results section to represent specific statements about results rather than types of analysis?

-       Lines 224-226: “In the case of splice variants, …” — I'm not sure we can draw this conclusion solely from Figure 2. More specific genes (in terms of Fig. 2a) might have more splice variants than the less specific genes, and all those variants might be increased in abundance proportionally. In this case, we would get similar distributions, but they would not say anything about 'splice specificity' per se. To support the conclusion made in the study, I think we should unfold Figure 2 and analyze how more specific splice variant for a specific gene relates to less specific splice variants of that gene. The fact that the purple and yellow circles in Figure 3 do not fully coincide partially accounts for this issue, but, as far as I understand, not fully.

Author Response

Authors used single splice variants and the degree of splicing as metrics for transcriptome data analysis. Using these metrics, they applied the gene set enrichment analysis to the data for three cell types (phenotypes) and demonstrated that these metrics can be informative to distinguish the transcriptome in these cell types, giving an additional information to the conventional gene expression levels.

I have the following comments/questions:

- Are there any other attempts to use the same metrics in the literature? The idea to analyze the abundances of splice variants instead of the total number of transcripts for a given gene seems straightforward given the technology, so it's hard to believe that this study is new regarding this aspect. If other attempts of this kind exist, they should be reviewed in the Introduction (which seems too short otherwise), and the novelty of the presented study should be specified more accurately.

We used two metrics to describe alternative splicing profiles: (1) the “degree of alternative splicing” defined as the number of detected splice variants per gene and (2) abundances (in TPM) of single splice variants. As to the first metric, we have suggested it very recently (ref. 16 of the revised manuscript) and, to the best of our knowledge, there was no other attempts to use this metric before. The second metric was used previously by other research groups but not for a particular case of selecting phenotype-specific transcript isoforms to differentiate between liver tissue and HepG2 and Huh7 cells. To address the Reviewer concern, we have made pertinent changes to the Introduction and Discussion sections (lines 73 - 81, 85 - 88, 92 - 96) as well as to the Summary and Abstract upon revision to clarify these issues and to specify the novelty of the presented study. Also, we have extended the Introduction by adding and shortly discussing 14 new references (refs. 10-15, 19-23, 27, 28, 33 of the revised manuscript), as a response to the Reviewer recommendation.   

- Is it possible to change the subsection titles in the Results section to represent specific statements about results rather than types of analysis?

The subsection titles in the Results section have been changed upon revision to make them more specific in regard to the presented results as suggested by the Reviewer.

-       Lines 224-226: “In the case of splice variants, …” — I'm not sure we can draw this conclusion solely from Figure 2. More specific genes (in terms of Fig. 2a) might have more splice variants than the less specific genes, and all those variants might be increased in abundance proportionally. In this case, we would get similar distributions, but they would not say anything about 'splice specificity' per se. To support the conclusion made in the study, I think we should unfold Figure 2 and analyze how more specific splice variant for a specific gene relates to less specific splice variants of that gene. The fact that the purple and yellow circles in Figure 3 do not fully coincide partially accounts for this issue, but, as far as I understand, not fully.

The conclusion in lines 224-226 (lines 232-235 of the revised manuscript) is simply states that the portion of phenotype-specific transcripts is higher than that of phenotype-specific genes – 50% vs. 30%. In fact, this conclusion was drawn not from Figure 2 but from the comparison of Tables S2 and S4 (the former contains 30556 ENST identifiers and 14427 ENSG identifiers while the latter – 15491 ENST identifiers and 4263 ENSG identifiers). We have stressed this point upon revision to avoid confusion.

As to whether more specific genes have more splice variants than the less specific ones, we attempted to look at the pertinent distributions, intrigued by the question raised by the Reviewer. We plotted histograms where each bar is the number of genes expressing the particular number of splice variants (the new Figure S1 of the Supplementary Material). We plotted such histograms for a set of all detected genes (14427), for two sets of genes which we randomly selected out of these 14427 genes (composed of 9235 and 4236 genes). The sizes of these two sets were chosen so as to match the sizes of genes with the phenotype-specific integral expression and with the phenotype-specific expression of one or two splice variants. It turned out that the distributions for the randomly selected genes are rather similar while the distributions for the specific sets of genes exhibit a notable decrease in the number of genes with a high level of alternative splicing. We added this observation to the Result section upon revision (lines 235-238).

All changes made in the manuscript upon revision are highlighted with yellow.

Round 2

Reviewer 2 Report

Comments and Suggestions for Authors

The authors have addressed all issues.